# Influence of Nb Addition on $\alpha''$ and $\omega$ Phase Stability and on Mechanical Properties in the Ti-12Mo-xNb Stoichiometric System

**Sinara Borborema [1,\*], Vitor de Holanda Ferrer [1], Adriana da Cunha Rocha [2], Caio Marcello Felbinger Azevedo Cossú [3], Aline Raquel Vieira Nunes [2], Carlos Angelo Nunes [3], Loic Malet [4] and Luiz Henrique de Almeida [2]**

1 Mechanical Department, Universidade do Estado do Rio de Janeiro—FAT/UERJ, Resende 27537-000, RJ, Brazil
2 Universidade Federal do Rio de Janeiro, Rio de Janeiro 21941-630, RJ, Brazil
3 Universidade de São Paulo—USP, Lorena 12612-550, SP, Brazil
4 Université Libre de Bruxelles—ULB, 1050 Brussels, Belgium
\* Correspondence: sinarab@msn.com

**Abstract:** Metastable β-Ti alloys have become one of the most attractive implant materials due to their high biocorrosion resistance, biocompatibility, and mechanical properties, including lower Young's modulus values. Mechanical properties of these alloys are strongly dependent on the final microstructure, which is controlled by thermomechanical treatment processing, in particular the Young's modulus and hardness. The aim of this work was to analyze the influence of phase precipitations in heat-treated Ti–12Mo–$x$Nb ($x = 0, 3, 8, 13, 17,$ and 20) alloys. The alloys were prepared via arc melting and treated at 950 °C/1 h, and then quenched in water. The microstructures were analyzed by optical microscopy, transmission electron microscopy, and X-ray diffraction. Mechanical properties were based on Vickers microhardness tests and Young's modulus measurements. Microstructural characterization showed that $\alpha''$ and $\omega$ stability is a function of Nb content for the Ti–12Mo base alloy. Nb addition resulted in the suppression of the $\alpha''$ phase and decrease in the $\omega$ phase volume fraction. Although the $\omega$ phase decreased with higher Nb contents, $\omega$ particles with ellipsoidal morphology were still observed in the Ti–12Mo–20Nb alloy. The $\alpha''$ phase suppression by Nb addition caused a marked increase in the Young's modulus, which decreased back to lower values with higher Nb concentrations. On other hand, the decrease in the $\omega$ phase continuously reduced alloy hardness. The study of the effect of chemical composition in controlling the volume fraction of these phases is an important step for the development of β-Ti alloys with functional properties.

**Keywords:** metallic biomaterials; microstructure; phases; titanium alloys; Young's modulus

## 1. Introduction

Elements considered safe, such as Nb, Ta, Zr, Mo, and Sn, are commonly selected for the development of new β-Ti alloys, as they are nontoxic and nonallergenic, and promote important characteristics for biomedical applications, such as excellent corrosion resistance, biocompatibility, and a better relationship between mechanical strength and Young's modulus. These properties vary according to the chemical composition of the β-Ti alloys, which can be classified as stable or metastable. [1–7].

Alloys classified as β-metastable are of greater interest due to the precipitation of second phases, such $\alpha'$, $\alpha''$, and $\omega$, when subjected to different thermomechanical cycles. The degree of β-metastability in titanium alloys can be controlled by applying the so-called molybdenum equivalent parameter (Mo$_{eq}$). If the Mo$_{eq}$ value is higher than 10 (%), metastable β is retained during quenching from temperatures above the β transus. Other phase transformations may occur during quenching, forming either the $\alpha'$ (hexagonal) or $\alpha''$ (orthorhombic) phases, depending on the concentrations of alloying elements. The metastable

ω phase (hcp structure) may also precipitate in the β-Ti matrix during quenching. As for the alloy, Young's modulus is mainly determined by the Young's modulus of individual phases; the addition of alloying elements and the combination of different thermomechanical treatments should be carefully performed in order to obtain a low modulus and high mechanical strength. From the literature, it is well known that among the phases normally formed in Ti-based alloys, the ω phase presents the highest elastic modulus and hardness [1–7]

In the study carried out by Bania [2] for alloys of the Ti–V–Cr–Al system, the parameter ($Mo_{eq}$) was proposed to identify the stability of the β phase in Ti alloys. For concentrations between 10 to 30 wt% $Mo_{eq}$, Ti alloys are classified as β-metastable. Xu et al. [3], observing the Ti–Mo–Cu system, mentioned that the presence of the metastable β phase will prevail with an 11% $Mo_{eq}$. Above 30 wt% $Mo_{eq}$, these alloys become β-stable. The $\alpha$ phase will be predominant in alloys with contents below 8 wt% $Mo_{eq}$. This parameter simplifies the determination of β phase stability in Ti alloys, predicting the microstructure according to their chemical composition.

Kalita et al. [4] showed that in Ti–Nb alloys the precipitation of ω is favored and, consequently, inhibits the precipitation of other phases, such as $\alpha$, $\alpha'$, and $\alpha''$, in the β-metastable matrix. Lauheurte et al. [5] studied the Ti–Nb–Ta–Zr system, analyzing the microstructural transformations at high temperatures for β-metastable alloys, based on the transformations of the β phase into second phases. These transformations occur by two processes: (i) nucleation and growth of the β-metastable phase and (ii) martensitic transformation, depending on the composition and cooling rate. The martensitic transformations (β → $\alpha''$) and (β → ω) cause different effects on the properties of metastable alloys. A study carried out by Xu et al. [6] on the Ti–Mo–Nb system showed that the decrease in Young's modulus and increase in mechanical strength of Ti alloys are controlled by thermomechanical processes. The formation of $\alpha''$ phase precipitates in the β-metastable matrix decreases the Young's modulus. The ω phase precipitates, on the other hand, increase the mechanical strength of these alloys.

This work aims to analyze the influence Nb addition on $\alpha''$ and ω phase stability, and on the mechanical properties and applicability as biomedical implants of the Ti–12Mo–$x$Nb alloys ($x$ = 0, 3, 8, 13, 17, and 20) after solution heat treatment at 950 °C for 1 h.

## 2. Materials and Methods

The Ti–12Mo–$x$Nb (wt%) ingots (40 g each) were melted from commercially pure Ti (grade 2), Nb, and Mo metals by arc melting with a tungsten electrode on a water-cooled copper hearth. The alloys were prepared under a high-purity argon atmosphere, and the ingots were melted six times to improve chemical homogeneity.

The ingots were heat treated at 950 °C for 1 h, and quenched in water using the procedure described by Gabriel et al. [7]. After heat treatment, no oxidation was observed on the surface of heat-treated parts.

The characterization of phases was performed using X-ray diffraction (XRD) operated at 40 kV and 30 mA with Cu–K$\alpha$ radiation ($\lambda$ = 1.5418 Å). The phases were identified via comparison with simulated diffractograms using the PHILIPS® X'Pert High Score 3.0 software with the PDF4–ICDD (PDF4–ICDD 3.0 plus, Malvern Panalytical, UK) database based on the JCPDS microfiches [8]. The volumetric fraction of the ω phase in the Ti-12Mo alloys with Nb addition was calculated using the peak fitting program of Origin with Pearson VII function. The integrated areas for ω diffraction peaks were analyzed.

The microstructure of the alloys was investigated by optical microscopy (OM). The sample was ground using silicon carbide papers up to 2400 mesh, polished using standard metallographic techniques, and then etched with Kroll's reagent (3 mL HF, 6 mL $HNO_3$, and 100 mL $H_2O$).

The microstructure of the alloys was also investigated by transmission electron microscopy (TEM) operated at 200 kV. The thin foils for this procedure were prepared by

twinjet electropolishing in a solution containing (60 mL) $HClO_4$, (590 mL) methanol, and (350 mL) ether monobutylethylene at 35 V and −20 °C.

Vickers microhardness values were determined using a durometer with a load of 100 gf for 20s. All hardness values were determined from the average of ten individual measurements [9,10].

The Young's modulus was obtained using an impulse excitation instrument ATCP® Sonelastic according to ASTM E1876–09. Samples had dimensions of 10 mm in diameter and 0.60 mm in thickness. Values for Young's moduli of the alloys were determined by the average of ten measurements [11]. Microhardness and elastic modulus of the Ti-6Al-4V ELI WQ alloy [12] in the solubilized condition were also used for comparison with those of the produced alloys.

### 3. Results and Discussion

Figure 1 shows the XRD patterns of heat-treated Ti–12Mo–$x$Nb alloys. The diffractogram shows the presence of ($\alpha''$ and $\omega$) martensitic phases in the β matrix for the Ti–12Mo alloy. With Nb addition, the transformation (β → $\alpha''$) was inhibited, with only the $\omega$ phase forming in matrix β, corroborating the results of Kalita et al. [6]. The increasing addition of Nb decreased the $\omega$ volumetric fraction of the studied alloys, amounting to 23% for Ti–12Mo–3Nb, 20% for Ti–12Mo–8Nb, 17% for Ti–12Mo–13Nb, and 13% for Ti–12Mo–17Nb. For the Ti–12Mo–20Nb alloy, the $\omega$ phase was not detected by X-ray diffraction, being observed only as small particles by TEM in the dark field mode.

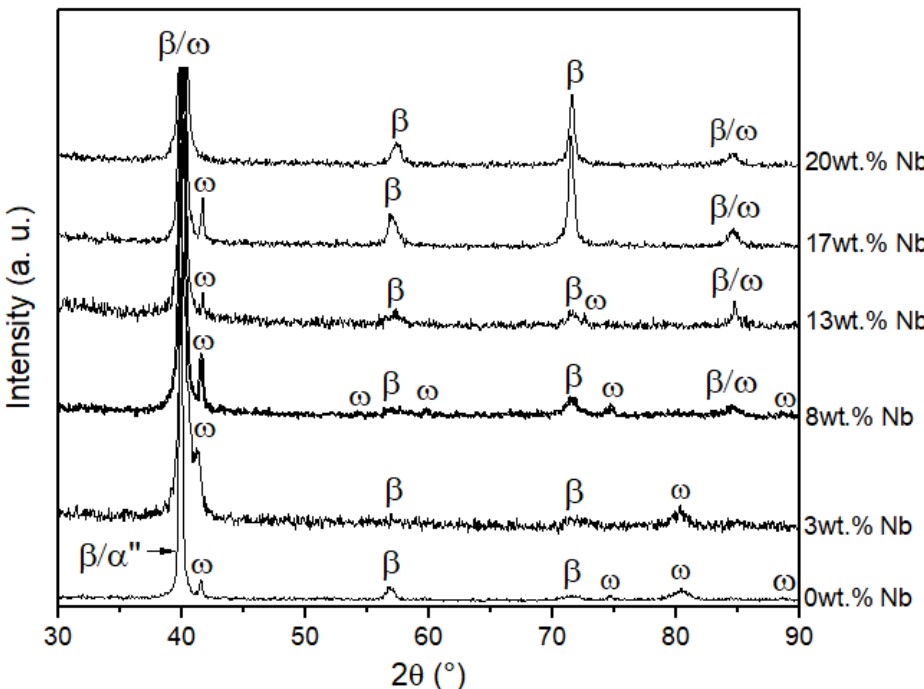

**Figure 1.** X-ray diffraction pattern of Ti–12Mo–$x$Nb ($x$ = 0, 3, 8, 13, 17, and 20) alloys treated at 950 °C/1 h. The diffractogram shows the presence of $\alpha''$ and $\omega$ martensitic phases in the β matrix for the Ti–12Mo alloy. With Nb addition, the $\alpha''$phase was inhibited, forming only the $\omega$ phase in matrix β.

Figure 2 shows optical micrographs of treated Ti–12Mo–$x$Nb alloys with different Nb additions. The heat treatment was not sufficient to eliminate the dendrite structures in the alloys. Considering the XRD results, it was found that the microstructure of the Ti–12Mo alloy (Figure 2a) consisted of grains of β phase with acicular $\alpha''$ precipitates. The presence of the $\alpha''$ phase in the Ti–12Mo alloy occurred in the interdendritic regions rich in Ti, suggesting a decrease in the concentration of β stabilizers in these regions. In agreement

with Raganya et al. [13], who showed that the interdendritic regions are richer in Ti and, consequently, poorer in β stabilizers, which can lead to the formation of the phases α′ and α″. The optical micrographs of Ti–12Mo–xNb (x = 3, 8, 13, 17, and 20) alloys showed only the β matrix, as shown, for example, in the alloy with 20 wt% Nb. Nb addition resulted in the suppression of the α″ phase. This is in accordance with studies conducted by Matsumoto et al. [14], which showed that the addition of Nb to the Ti alloys causes the suppression of the α″phase. In Ti–12Mo–xNb alloys, it was not possible to identify the ω phase by OM, due to its size (nanometer scale).

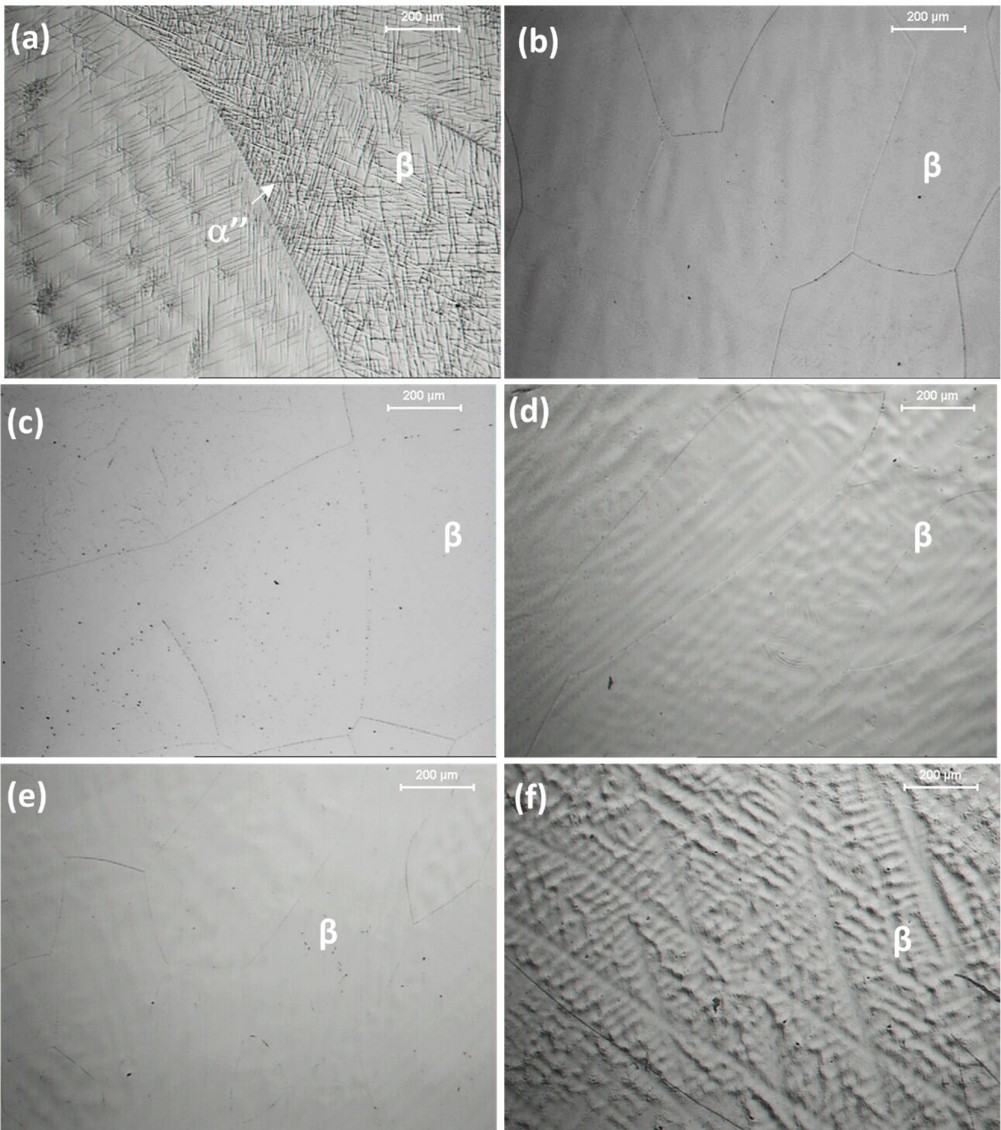

**Figure 2.** Optical microscopy of Ti–12Mo–xNb alloys treated at 950 °C for 1 h: (**a**) x = 0; (**b**) x = 3; (**c**) x = 8; (**d**) x = 13; (**e**) x = 17; (**f**) x = 20.

Figure 3 shows the images obtained by TEM of the Ti–12Mo, Ti–12Mo–3Nb, and Ti–12Mo–20Nb alloys in the 950 °C heat treatment condition for 1 h. Figure 3a shows the dark-field image of the Ti–12Mo alloy, where many ω phase precipitates are observed with an ellipsoidal morphology, in a β phase matrix. Figure 3b shows the brightfield image of the Ti–12Mo alloy, where the presence of the α″ phase in the β matrix is observed, corroborating the identification of the α″ phase in the X-ray spectrum, presented in Figure 1. Figure

3c,d show dark-field images of the of Ti–12Mo–3Nb and Ti–12Mo–20Nb alloys, respectively, where only ω phase precipitates can be seen. The Nb addition resulted in the suppression of the $\alpha''$ phase and decrease in the ω phase, confirming the effect of $Mo_{eq}$ on β stabilizing. Although the amount of ω phase decreases with higher Nb contents, ω precipitates with ellipsoidal morphologies can be observed on the thin foil of the Ti–12Mo–20Nb alloy.

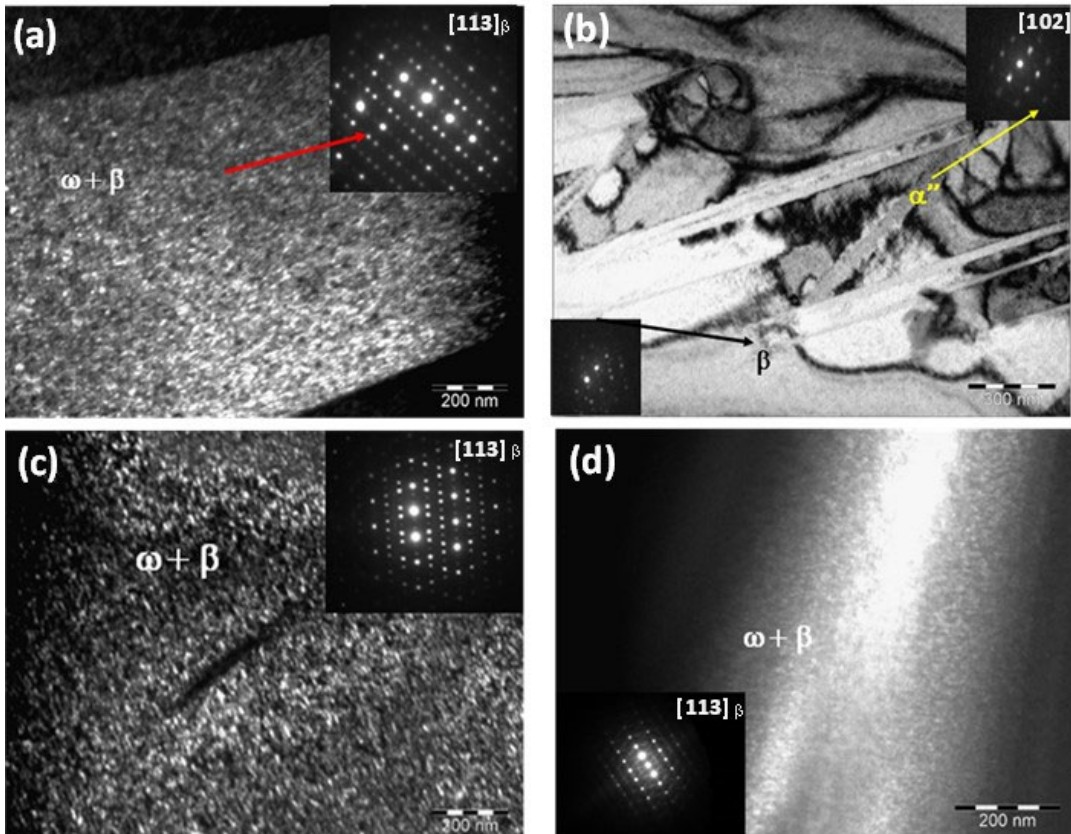

**Figure 3.** (**a**) TEM dark-field image of ω phase with an ellipsoidal morphology precipitated in a β matrix in the Ti–12Mo alloy and SAED pattern of β- and ω-Ti with the [113]β axis zone; (**b**) bright-field image of the martensitic $\alpha''$ lamellae and SAED pattern of β and $\alpha''$ phases in the Ti–12Mo alloy; (**c**) and (**d**) dark-field images of the ω precipitates with an ellipsoidal morphology in a β matrix in the Ti–12Mo–3Nb and Ti–12Mo–20Nb alloys, respectively.

Table 1 shows the values of Vickers hardness, Young's modulus, and, the hardness to modulus ratio for Ti–12Mo–xNb alloys heat treated at 950 °C for 1 h. The hardness values ranged from 224 to 449 HV, highlighting the Ti–12Mo alloy (449 HV), and the lowest hardness values were obtained for the Ti–12Mo–17Nb and Ti–12Mo–20Nb alloys (224 HV).

Young's modulus values ranged from 59 to 81 GPa, highlighting the Ti–12Mo–20Nb alloy (59 GPa). In this condition, just a small amount of ω phase precipitates were observed in the β matrix, thus showing no influence on the mechanical properties. The highest Young's modulus values were obtained for the Ti–12Mo–xNb alloys (x = 3, 8, and 13), which may be related to the high volume fraction of phase ω in the β matrix.

Yao et al. [15] stated that the presence of the $\alpha''$ phase has a lower Young's modulus than the β phase. Matsumoto et al. [14] reported that phase ω presents higher modulus values than $\alpha''$ and β phases. The Ti–12Mo alloy, despite having the ω phase in the β matrix, presented a low Young's modulus (60 GPa), due to the precipitation of the $\alpha''$ phase in the β matrix. In Ti–12Mo–xNb alloys, hardness decreases with increasing Nb concentrations, which is related to the decrease in the ω phase precipitation in the β matrix.

Considering that hardness has a direct relation with mechanical strength [16,17], to estimate the performance of biomaterials to be used as bone substitute, the hardness to Young's modulus ratio was calculated [18–20]. The higher the value resulting from this ratio, the greater the potential for use as bioimplants. According to Table 1, all the alloys presented higher hardness to Young's modulus ratio than the traditional Ti-6Al-4V alloy; amongst these, the highest ratio was obtained for the Ti–12Mo alloy (7.5).

**Table 1.** Properties and phases of Ti–12Mo–xNb alloys treated at 950 °C/1 h and, for comparison, solubilized Ti–6Al–4V ELI alloys data from [11].

| Alloys | Hardness (HV) | Young's Modulus (GPa) | Phases | Ratio (Hardness/Modulus) |
|---|---|---|---|---|
| Ti-6Al-4V ELI WQ | 346 | 118 | $\alpha + \beta$ | 2.9 |
| Ti–12Mo | $449 \pm 4$ | $60.0 \pm 1.8$ | $\alpha'' + \omega + \beta$ | 7.5 |
| Ti–12Mo–3Nb | $380 \pm 10$ | $81.0 \pm 0.3$ | $\omega + \beta$ | 4.7 |
| Ti–12Mo–7Nb | $329 \pm 3$ | $80.0 \pm 1.5$ | $\omega + \beta$ | 4.1 |
| Ti–12Mo–13Nb | $260 \pm 5$ | $76.0 \pm 4.9$ | $\omega + \beta$ | 3.4 |
| Ti–12Mo–17Nb | $224 \pm 8$ | $59.0 \pm 0.4$ | $\omega + \beta$ | 3.5 |
| Ti–12Mo–20Nb | $224 \pm 2$ | $63.0 \pm 0.2$ | $\omega + \beta$ | 3.8 |

Figure 4 shows the Vickers microhardness and modulus values as functions of Nb content. For higher Nb concentration, the hardness decreases continuously because of the $\beta$ phase stabilization and, at the same time, due to the smaller amount of $\omega$ precipitates.

The Young's modulus value showed an increase when Nb was added compared with the Ti–12Mo alloy. This behavior can be explained due to the $\alpha''$ suppression as a consequence of $\beta$ stabilization promoted by Nb addition. This is in accordance to the literature that states that Young's moduli of different phases have the given order: $E\alpha'' < E\beta < E\alpha < \omega$ [21–23]. Young's moduli of multiphase materials are determined by the present phases and the individual Young's modulus of each phase [24].

From this point on, Young's modulus showed a similar behavior to that of the hardness as a consequence of the reduction in $\omega$ precipitation.

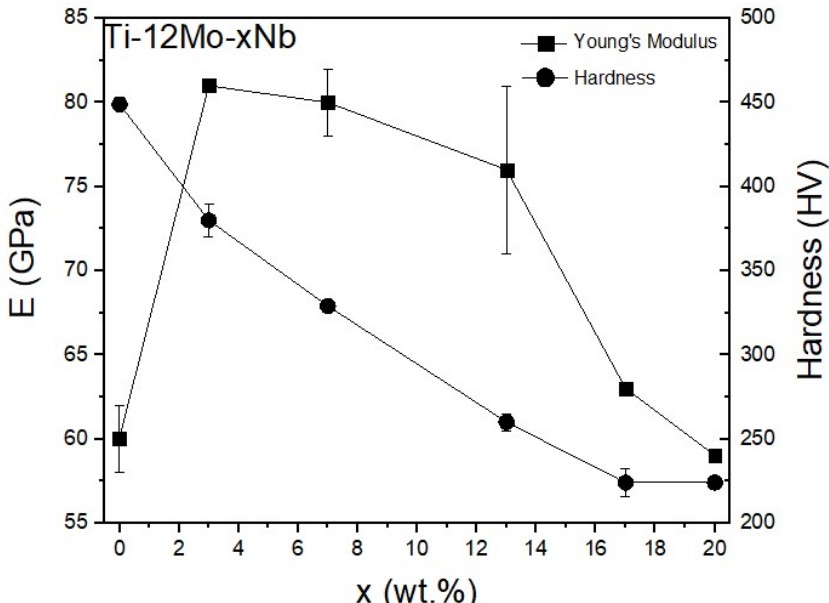

**Figure 4.** Microhardness and Young's modulus values in function of addition Nb.

## 4. Conclusions

Microstructural characterization results showed the presence of $\alpha''$ and $\omega$ in the $\beta$ matrix of the Ti–12Mo alloy.

Higher concentrations of added Nb resulted in the suppression of the $\alpha''$ phase and the decrease in the $\omega$ phase, confirming the effect of Mo$_{eq}$ on $\beta$ stabilization.

Young's Moduli values showed an increase, when compared to those of the Ti–12Mo alloy, when Nb was added having as a consequence $\alpha''$suppression. For the other alloys, increases in Nb concentration present a direct relationship between hardness and Young's modulus, as a consequence of the reduction in $\omega$ precipitation.

**Author Contributions:** Conceptualization, S.B.; methodology, V.d.H.F., C.M.F.A.C.; validation, A.d.C.R., A.R.V.N. and C.A.N.; formal analysis, L.M. and A.d.C.R. and S.B.; investigation, S.B.; writing—original draft preparation S.B.; writing—review and editing, L.H.d.A.; funding acquisition, S.B. and L.H.d.A. All authors have read and agreed to the published version of the manuscript."

**Funding:** This research was funded by FAPERJ, grant number E26/202654/2019, CNPq, grant PQ-1B 12/2017 and PROSCIENCE/UERJ/FAPERJ.

**Data Availability Statement:** Not applicable.

**Acknowledgments:** This work was supported by CNPq (PQ–Universal), FAPERJ (CNE), and CAPES.

**Conflicts of Interest:** The authors declare no conflict of interest.

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
