# Peer review of "Influence of Nb Addition on α″ and ω Phase Stability and on Mechanical Properties in the Ti-12Mo-xNb Stoichiometric System"

_metals, doi:10.3390/met12091508_

Round 1

Reviewer 1 Report

The paper “Influence of Nb Addition on α’’ and ω Phase Stability and on 2 Mechanical Properties in the Ti-12Mo-xNb Stoichiometric 3 System” investigates the effect of Nb addition into the Ti12Mo alloy on the microstructure and mechanical properties. Alloys were produced by arc melting and then a water quenching treatment is achieved. The paper describes and discusses the results obtained based on the β-Ti phase stabilization, affirming that α’’-Ti phase is suppressed by the effect of Nb addition. This changes in microstructure generated an increment on the Young’s modulus and reduction on the hardness.

The paper is well structured, but with some language errors and some obvious mistakes that should be improved. The major concern in the paper is that all the discussion is supported by the presence of the α’’ and ω phases, however, there are not a quantification of such phases in order to give a clear scenario to understand the effect on mechanical properties. Thus, it will be necessary to quantify each phase from the X-ray patterns. otherwise, the discussion doesn't have scientific support.

The second major issue that authors should addressed is to explain and discuss why the Young modulus increased and the Hardness decreased, since as it is pointed out in the paper, these two properties show a direct relationship, which means that if one of them increased the other should also increase. But the results show the contrary, why?

Introduction should be improved with more literature of the martensitic phases obtained by the addition of Nb, Ta, Zr into the Ti. Some papers dealing with this subject are listed in here but there are a lot of information about the subject.

Hao, Y. L., Yang, R., Niinomi, M., Kuroda, D., Zhou, Y. L., Fukunaga, K., & Suzuki, A. (2002). Young’s modulus and mechanical properties of Ti-29Nb-13Ta-4.6 Zr in relation to α ″martensite. Metallurgical and Materials Transactions A, 33(10), 3137-3144.

Hao, Y. L., Yang, R., Niinomi, M., Kuroda, D., Zhou, Y. L., Fukunaga, K., & Suzuki, A. (2003). Aging response of the Young’s modulus and mechanical properties of Ti-29Nb-13Ta-4.6 Zr for biomedical applications. Metallurgical and Materials Transactions A, 34(4), 1007-1012.

Zhao, G. H., Liang, X. Z., Kim, B., & Rivera-Díaz-del-Castillo, P. E. J. (2019). Modelling strengthening mechanisms in beta-type Ti alloys. Materials Science and Engineering: A, 756, 156-160.

Ozaki, T., Matsumoto, H., Watanabe, S., & Hanada, S. (2004). Beta Ti alloys with low Young's modulus. Materials transactions, 45(8), 2776-2779.

Chen, L. Y., Cui, Y. W., & Zhang, L. C. (2020). Recent development in beta titanium alloys for biomedical applications. Metals, 10(9), 1139.

Haghighi, S. E., Lu, H. B., Jian, G. Y., Cao, G. H., Habibi, D., & Zhang, L. C. (2015). Effect of α ″martensite on the microstructure and mechanical properties of beta-type Ti–Fe–Ta alloys. Materials & Design, 76, 47-54.

Motyka, M. (2021). Martensite formation and decomposition during traditional and AM processing of two-phase titanium alloys—An overview. Metals, 11(3), 481.

Yang, Y., Castany, P., Bertrand, E., Cornen, M., Lin, J. X., & Gloriant, T. (2018). Stress release-induced interfacial twin boundary ω phase formation in a β type Ti-based single crystal displaying stress-induced α” martensitic transformation. Acta Materialia, 149, 97-107.

Minor corrections:

Improve quality of Fig. 1

I suggest you to carefully check the whole paper to eliminate grammatical or spelling errors.

Improve conclusions highlighting the main finding in this work avoiding to do a summary of the discussion.

My recommendation is to be accepted with major corrections as the ones describe above.

Author Response

Attached clarifications 

Reviewer 2 Report

The manuscript “Influence of Nb addition on α’’ and ω phase stability and on mechanical properties in the Ti-12Mo-xNb stoichiometric system” is devoted to the study of the effect of phase precipitation in heat-treated Ti-12Mo-xNb alloys (x = 0, 3, 8, 13, 17 and 20) on the microstructure and mechanical properties. The authors made a detailed phase analysis and determined the values of the Young's modulus and microhardness of the alloys under study. The results obtained were analyzed and appropriate conclusions were presented.

I have the following comments on the manuscript:

1) Materials and Methods. “The hardness and Young's modulus of ASTM F136 Ti–6Al–4V wrought annealed alloy were also replicated for comparison under the same conditions described above. The values microhardness and elastic modulus of commercially Ti-6Al-4V alloy were also determined for comparison with those of the produced alloys.” Why do the authors compare the properties of the investigated alloys in the state of casting and annealing with the deformed Ti–6Al–4V alloy? I think it is incorrect to compare the properties of alloys in different states of processing.

2) Table 1. E = 63 ± 0 GPa, 59 ± 0 GPa and 81 ± 0 GPa. In the “Materials and Methods” the authors write that “The value of the Young’s modulus of the alloys was determined by the average of ten measurements”. So how can the confidence interval be 0 in this case?

3) Results and Discussion. Figure 1. Lines 99-100. “The diffractogram showed the presence of (α'' and ω) martensitic phases in β matrix for Ti-12Mo alloy”. As can be seen from the X-ray patterns of the alloys (Figure 1), the peaks of the α'' and ω martensitic phases correspond to the same angle. How did the authors determine the disappearance of the ω phase in the alloys from the X-ray pattern?

Author Response

Attached clarifications 

Round 2

Reviewer 1 Report

The authors improved the paper according to the suggestions.

Reviewer 2 Report

The authors significantly revised the manuscript and answered all questions and comments. I think the manuscript can be accepted for publication.